# Cost-Effectiveness of Two Dry Needling Interventions for Plantar Heel Pain: A Secondary Analysis of an RCT

**DOI:** 10.3390/ijerph18041777

**Published:** 2021-02-12

**Authors:** Daniel Fernández, Zaid Al-Boloushi, Pablo Bellosta-López, Pablo Herrero, Manuel Gómez, Sandra Calvo

**Affiliations:** 1Faculty of Health Sciences, Universidad San Jorge, Campus Universitario, Autov. A23 km 299, Villanueva de Gállego, 50830 Zaragoza, Spain; efernandez@usj.es (D.F.); pbellosta@usj.es (P.B.-L.); agomez@usj.es (M.G.); scalvo@usj.es (S.C.); 2Ministry of Health, State of Kuwait, Jamal Abdulnasser Street, Al Solaibeykhat Area 5, Safat, Kuwait City 13001, Kuwait; boloushi@gmail.com; 3Physiatry and Nursing Department, Faculty of Health Sciences, Zaragoza University, C/Domingo Miral s/n, CP 50009 Zaragoza, Spain

**Keywords:** dry needling, percutaneous needle electrolysis, cost-effectiveness analysis, plantar heel pain, quality of life

## Abstract

Plantar heel pain is a common cause of foot pain that affects patients’ quality of life and represents a significant cost for the healthcare system. Dry needling and percutaneous needle electrolysis are two minimally invasive treatments that were shown to be effective for the management of plantar heel pain. The aim of our study was to compare these two treatments in terms of health and economic consequences based on the results of a published randomized controlled trial. For this, we evaluated the costs from the point of view of the hospital and we carried out a cost-effectiveness study using quality of life as the main variable according to the Eq-5D-5L questionnaire. The cost of the complete treatment with dry needling (DN) was €178.86, while the percutaneous needle electrolysis (PNE) was €200.90. The quality of life of patients improved and was translated into +0.615 quality-adjusted life years (QALYs) for DN and +0.669 for PNE. PNE presented an average incremental cost-effectiveness ratio (ICER) of €411.34/QALY against DN. These results indicate that PNE had a better cost-effectiveness ratio for the treatment of plantar heel pain than DN.

## 1. Introduction

Plantar heel pain (PHP) is a common cause of foot pain and discomfort that affects the quality of life (QoL) of patients. The frequency and incidence of PHP is uncertain; however, it is estimated that over the course of a lifetime, 10% of the population might suffer from this condition [1,2]. PHP places a relatively large burden on public health, being a fairly common cause of outpatient visits [3] and medical expenses, with cost of PHP treatment estimated to be between $192 and $376 million in the United States [4].

Clinical practice guidelines recommend the use of conservative treatment for PHP management, such as joint and soft tissue mobilization or self-stretching home programmes [5]. In particular, self- stretching home programmes were shown to be effective for addressing PHP [6]. Furthermore, recent randomised clinical trials (RCTs) showed that there is an additional effect of reduction of pain severity when self- stretching home programmes are combined with manual therapy [7]. It was also shown that dry needling (DN) is effective for managing PHP, although the magnitude of the treatment effect should be considered against the frequency of minor transitory adverse events [8].

DN and percutaneous needle electrolysis (PNE) are two minimally invasive treatment modalities that were shown to be effective for PHP management, reducing the mean and maximum pain since the first treatment session, with long lasting effects [9]. Despite the increased use of these treatment modalities in recent years in clinical practice, to the best of our knowledge, no previous study investigated the cost-effectiveness of PNE and DN on PHP. Moreover, in the case of PHP, we only found a study analysing the cost-effectiveness of other interventions, such as the use of foot orthosis [10].

Therefore, the aim of this study was to compare the cost-effectiveness of two alternative interventions, DN and PNE, in terms of health and economic consequences.

## 2. Materials and Methods

### 2.1. Study Design

The primary study was a randomised controlled trial carried out during 2018 and 2019, to compare the effectiveness of two needling interventions, conducted at the Physical Medicine and Rehabilitation Hospital in Kuwait. The study was conducted in compliance with the Helsinki Declaration of Human Rights and ethical approval was obtained from the Medical Ethics Committee of the State of Kuwait Ministry of Health, with reference number 642/2017 and registration at Clinicaltrials.com, number NCT03236779, available at https://pubmed.ncbi.nlm.nih.gov/30683124/ (accessed on 15 December 2020).

This secondary study used the effectiveness data obtained in the RCT to perform a cost-effectiveness study.

### 2.2. Participants

Participants were included in the randomised controlled trial if they fulfilled the following criteria. (1) Diagnosed with PHP in accordance with the Clinical Guidelines linked to the International Classification of Function, Disability and Health from the Orthopaedic Section of the American Physical Therapy Association. (2) Aged 21 to 60 years at admission to the study, according to the Kuwaiti Ethical Committee. (3) A history of PHP for over one month, showing no improvements with previous conservative treatment. (4) The ability to walk 50 m without any support. (5) The presence of myofascial trigger points on plantar and calf muscles based on an initial physical examination carried out by a physiotherapist with experience and training in myofascial trigger points. (6) Accepting treatment from a male physiotherapist. (7) The ability to understand the study and give informed consent, as well as having signed the consent form. The exclusion criteria were—(1) needle phobia; (2) needle allergy or hypersensitivity to metals; (3) the presence of coagulopathy or use of anticoagulants according to medical criteria; (4) the presence of peripheral arterial vascular disease; (5) pregnancy; (6) dermatological disease affecting the dry needling area; (7) the presence of any chronic medical condition that might preclude participation in the study, such as malignancy, systemic inflammatory disorders (e.g., rheumatoid arthritis, psoriatic arthritis, ankylosing spondylitis, septic arthritis), neurological diseases, polyneuropathy, mononeuropathy, and sciatica; (8) treatment of PHP with needling or acupuncture during the last four weeks; (9) history of injection therapy in the heel over the previous three months; and (10) previous history of foot surgery or fracture.

### 2.3. Intervention Conditions

The intervention conditions were established during the RCT. All participants signed the informed consent and were randomised to the DN or PNE group in blocks of 10 patients. Allocation was randomly assigned using a computer program (Randomizer, https://www.randomizer.org/ (accessed on 14 January 2018)) with random patient file number sequences generated by a third person not involved in the study. Each participant received four individual physical therapy sessions consisting of DN or PNE, once a week, using specific DN needles (Agu-punt, Barcelona, Spain) [9]. The duration of each session was approximately 30 min. DN and PNE interventions were performed exactly in the same way, with the only difference being that the needle was connected to the PNE equipment during the PNE procedure (“Physio Invasiva” equipment PRIM S.A. Madrid, Spain) to add a galvanic current to the mechanical effect of the needle. During the first session, all patients were instructed in a self-stretching protocol [7] that was performed twice a day. An independent blinded assessor conducted all assessments at baseline, 4, 8, 12, 26, and 52 weeks of follow-up. Quality of life (QoL) data were used to perform the cost-effectiveness analysis.

### 2.4. Costs

The economic evaluation was made retrospectively and from a hospital perspective. From this, direct healthcare costs were calculated, such as the material used to perform the needling (sterile gauze, disinfectant, needles), PNE equipment, and the cost of the physiotherapy and physiatrist sessions. To assess the cost of PNE equipment, two scenarios were calculated. In one scenario, the total cost of the equipment was included in the cost of the study, calculating the total number of sessions that were carried out in our study and the total cost of the equipment was divided by the number of sessions. In the second scenario, the average number of uses that the equipment can have in a standard clinic per year and the amortization of the cost of the equipment in a short period of three years was calculated.

The cost of the physiatrist and the physical therapy was obtained from the bulletin “Cost analysis and performance evaluation for government health services of the Kuwait Ministry of Health” (Financial affairs, Budget and Control Department—Cost Accounting Section) [11]. Costs of usual care were not considered for comparison as the participants selected were enrolled on the basis that the standard treatment for PHP was not effective (see inclusion criteria).

### 2.5. Outcomes

EQ-5D-5L was completed at baseline, 4, 8, 12, 26, and 52 weeks for each patient. This questionnaire included five dimensions—mobility, self-care, usual activities, pain/discomfort and anxiety/depression, with five levels of severity, which made it even more robust than previous versions [12]. To obtain the quality-of-life value based on the EQ-5D questionnaire data, the ‘EQ-5D-5L Index Value Crosswalk’ tool was used with English preference weights. Using the study timing, it was possible to obtain the quality-adjusted life years (QALYs), which was the preferred measure of health outcomes for use in technology appraisals. QALYs were estimated for each subject using area under the curve analysis.

### 2.6. Statistical Analysis

The statistical analysis was performed using IBM SPSS Statistics (V.25, IBM, Madrid, Spain) by intention to treat, with the last observation carried forward. The distribution of the data was verified by the Kolmogorov-Smirnov normality test and a factorial ANOVA (Bonferroni test) was performed to evaluate the differences.

To analyze the uncertainty of QALY and costs, a probabilistic analysis was performed by varying both parameters according to certain ranges of their distribution variable—QALY and cost pairs were bootstrapped 1000 times to achieve 95% confidence intervals [13]. To summarize, this study considered the proportion of bootstrap replications that fell below and to the right of the line of the cost-effectiveness threshold. Then, we represented the incremental cost-effectiveness ratio (ICER) in relation to possible values of the cost-effectiveness threshold in the cost-effectiveness acceptability curve (CEAC) [14]. To complete this sensitivity analysis, a univariate analysis of overhead costs was carried out considering that they could account for up to 30% of the value of direct costs [15].

## 3. Results

A total of 102 patients with an average age of 48 years were included in the final economic analysis. Participants were divided in two groups, without significant differences between groups at baseline (Table 1).

All results from the original study titled “Comparing two dry needling interventions for plantar heel pain: a randomised controlled trial” can be found at https://bmjopen.bmj.com/content/bmjopen/10/8/e038033.full.pdf.

### 3.1. Costs

The DN and PNE treatment costs evaluated are displayed in Table 2. Costs were not compared with the usual standard of care costs as one of the inclusion criteria to be enrolled in the study was “a history of PHP for over one month, showing no improvements with previous conservative treatment”.

The cost of sanitary consumables corresponded to the use of gloves, needles, disinfectant, and any other material associated with the needling interventions. The cost of the PNE equipment corresponded to the price of the “Physio Invasiva” equipment (PRIM S.A. Madrid, Spain), without taxes. With these data, we calculated the cost per session and asked for feedback from a group of expert physical therapists to closely resemble the real value of the cost of a session with PNE in the usual clinical scenario.

To complete the costs, it was necessary to add the cost of the personnel who carried out the treatments and their supervision. These costs are available in the document “Cost analysis and performance evaluation for government health services 2013–2014” from the Kuwait Ministry of Health [11]. The cost of the physiatrist and the physical therapy was considered to be the same in both groups. Total direct costs of treatment were €22.04 more expensive in the PNE than in the DN group, because of the use of the specific equipment for providing the PNE treatment.

### 3.2. Quality of Life

An improvement in QoL was observed at weeks 4 and 8 in both treatment groups. However, the PNE group also showed improvements at 52 weeks (Table 3)*,* with significant differences between groups at 52 weeks (Figure 1).

Based on the obtained QoL results, treatment with DN presented an improvement of +0.615 QALYs while PNE had an improvement of +0.669 (difference of 0.054) QALYs. The probability of each alternative being cost effective was reviewed using the bootstrap technique as a probability sensitivity analysis.

The cost-effectiveness ratios of PNE treatment versus DN increased during the study period (Table 4).

PNE showed a clear dominance against DN at 52 weeks, presenting even higher costs for the equipment used (Figure 2)

The probability that the PNE treatment at 52 weeks would have a better cost-effectiveness ratio compared to DN was 86%.

The threshold that was used to evaluate the implementation of a therapeutic alternative is usually €25,000. However, in our study, PNE presented an 86% probability of being more cost-effective against DN at values much lower than this threshold (Figure 3). Moreover, PNE presented an average ICER of €411.34/QALY against DN.

The univariate analysis of estimated overhead costs implied an increased cost of €60.27 for the PNE group (total €261.17) and €53.66 for the DN group (total €232.52). The resulting ICER was €530.55/QALY, 119.25€/QALY more than when overhead costs were not considered.

## 4. Discussion

The objective of this study was to analyze which treatment option was more cost-effective for PHP treatment, on the basis that both DN and PNE were demonstrated to be effective. For this, we considered that a cost-effectiveness study was the best tool, using a good efficacy marker such as the patient QoL and evaluating all costs involved at the health centre. The baseline characteristics of our patient sample were homogeneous, and it was a good starting point to exclusively assess the variation due to treatment.

We chose the EQ-5D-5l scale mainly because it was one of the most used scales, but also because it allowed us to make a more global assessment of patient health status. Comparisons of EQ-5D-5l with other more specific scales such as the Manchester Oxford Foot Questionnaire (MOXFQ) concluded that both were particularly responsive to changes in pain, mobility and activity, or social interaction following treatment; but ‘the generic EQ-5D may allow comparison of general health states in the wider health community’ [16]. However, it has a drawback when it comes to transferring the data from the questionnaire to QoL, as currently, there is no predefined crosswalk index for Kuwaiti patients. In communication with EuroQol group experts, it was considered that the most valid option could be to use the British values.

Our analysis showed an improvement of the QoL in both groups of almost 0.1 point on the EQ-5D-5l scale. The results obtained by Rome et al. [10] with the use of an accommodative foot orthosis or functional foot orthosis were 0.07 and 0.09 QALY/year, respectively. If we compare the QALY values gained with DN treatment (0.615) and PNE (0.669) against these results, we see that there was a considerable improvement with these needling treatments.

Regarding costs, only those that most influence a clinic, hospital, or a health centre were considered in terms of equipment and staff. In both cases, the cost of the equipment was assessed as the consumable material that was used and renewed for each patient and session. The price of the PNE equipment might have differences depending on the brand or the country where it was purchased. In this case, we only valued the device used in this study. To assess the cost of this equipment for each session as accurately as possible, an average of uses per year and years of amortization was determined. On the other hand, we also calculated the cost involved for each session in case the equipment was used only for this study (51 patients, 4 sessions for each one; 204 sessions). The cost with the average of annual uses and amortization seemed to be the most realistic since this equipment could be used in many other pathologies.

On the contrary, being among the first to perform cost analyses of this type of techniques might lead to certain limitations, such as the difficulty of assessing the indirect costs. In this study, we used as a reference the inclusion of indirect costs, represented by a large weight in the overall evaluation of costs, which might bias the results [17]. The use of RCT data retrospectively did not allow us to obtain more information of indirect costs and therefore we did not include them to avoid the distortion of results.

There were improvements in QoL with both treatments, and the differences in costs between treatments were relatively low. The cost-effectiveness relationships analyzed after the bootstrapping indicated that there was a clear advantage of PNE over DN at 52 weeks. These results must be interpreted with caution, taking into account the methodological limitations of our study, mainly related to the sample size, the lack of studies that served as a comparison and the high rate of drop-outs. In the latter case, and in the same way as explained in the published study “either underpowering or the intention to treat analysis may explain the inconsistency of the results in the PNE group possibly leading to significant results in weeks 12 and 26” [9], also in cost-effectiveness ratio.

However, this study also had some strengths, as it is the first study to follow a strict methodology in the cost-effectiveness analysis of two needling interventions and the data were extracted from a randomized controlled trial comparing these needling interventions.

This was the first time that a study compared the treatment costs of DN and PNE, which combined with the cost-effectiveness analysis carried out. This could be helpful for both administrators and clinicians to assess the inclusion of new treatment methods in hospitals or rehabilitation clinics.

## 5. Conclusions

Although both DN and PNE were found to be effective in improving QoL for PHP management, PNE treatment was more cost-effective than DN, with significant differences at 52 weeks. In the comparisons made according to the cost-effectiveness analysis, this translated into an 86% probability that PNE was more cost-effective compared to DN at 52 weeks. Considering the limitations of our study based on the loss of patients from the previous study and the lack of reliable indirect costs, further studies are needed to verify the differences in PNE and DN.

## Figures and Tables

**Figure 1 ijerph-18-01777-f001:**
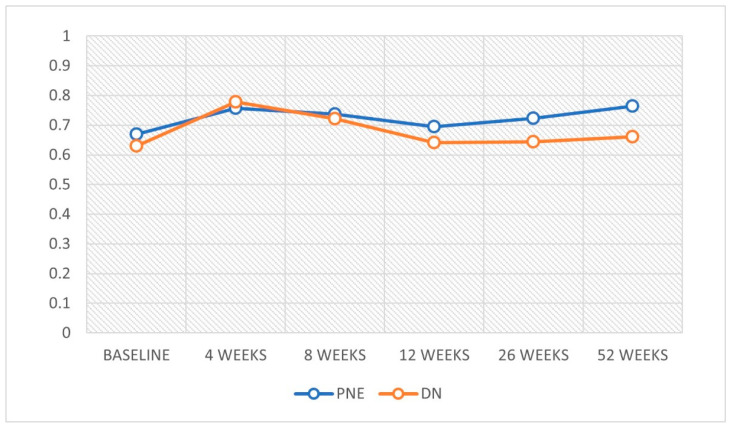
Variation in quality of life throughout the study.

**Figure 2 ijerph-18-01777-f002:**
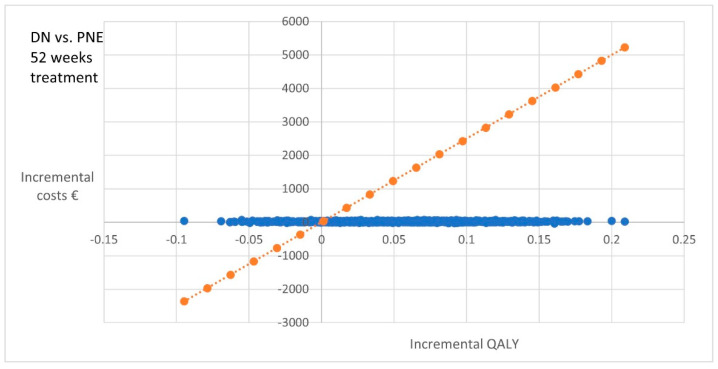
Bootstrapped cost-utility pairs dry needling (DN) vs. percutaneous needle electrolysis (PNE).

**Figure 3 ijerph-18-01777-f003:**
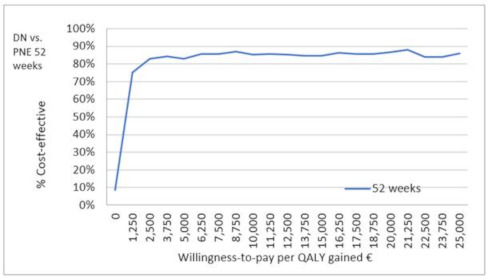
Cost-effectiveness acceptability curve (CEAC).

**Table 1 ijerph-18-01777-t001:** Demographic characteristics of participants and other baseline characteristics.

	DN Group N = 51	PNE Group N = 51
Gender (male)	15 (29.4%)	15 (29.4%)
Age (years)	49.5 (±8.9)	48.1 (±8.8)
Height, cm	160.5 (8.2)	161.2 (7.9)
Weight, kg	87.5 (16.5)	90.8 (15.2)
Body mass index (kg/m^2^)	33.9 (±5.5)	35.1 (±6.4)
Duration of symptoms, months	6.0 (6.0)	9.9 (11.5)
Medications, n yes (%)		
- Neuromodulators/antiepileptic	18 (35.3)	22 (43.1)
- Painkillers	16 (31.4)	16 (31.4)
- Anti-inflammatory medication	16 (31.4)	17 (33.3)
- Myorelaxant medication	9 (17.6)	8 (15.7)

All values are means (± standard deviation) or number of patients (%).

**Table 2 ijerph-18-01777-t002:** Costs.

	DN Group	Both	PNE Group
Sanitary consumables (session)	€1.06		
PNE equipment	-		€2300
Cost of PNE per session (204 sessions)			€12.21
Cost of PNE per session according to clinic			€0.94
Physiatrist (session)	€96.03		
Physical therapy (session)	€43.65		
Cost first week of treatment	€140.74		€146.26 ± €5.64 *
Total (four sessions)	€178.86		€200.90 ± €22.55

* The variations in the cost for the PNE group are due to the calculation with or without amortization of the PNE equipment.

**Table 3 ijerph-18-01777-t003:** Mean quality of life (QoL) from EQ-5d.

Variable	DN Mean (SD)	DN Mean Difference from Baseline	PNE Mean (SD)	PNE Mean Difference from Baseline	Mean Qol Difference PNE–DN (SD)
**QoL (1-0)**					
Baseline	0.630 (0.230)	-	0.669 (0.215)	-	0.039 (0.323)
4th week	0.777 * (0.224)	0.147	0.757 * (0.237)	0.087	−0.020 (0.341)
8th week	0.721 * (0.232)	0.091	0.737 * (0.230)	0.067	0.015 (0.349)
12th week	0.641 (0.298)	0.011	0.694 (0.272)	0.025	0.053 (0.370)
26th week	0.644 (0.287)	0.014	0.722 (0.278)	0.053	0.078 (0.365)
52nd week	0.660 (0.270)	0.034	0.764 * (0.099)	0.099	0.103 ** (0.323)

(± standard deviation); * difference from baseline in means is significant *p*-value < 0.05; ** difference between groups is significant.

**Table 4 ijerph-18-01777-t004:** Variation of cost-effective ratio during the study.

	PNE				
Weeks	4	8	12	26	52
% Cost-effective	50%	46%	63%	79%	86%

## Data Availability

Data will be available on reasonable request by email to Daniel Fernández (efernandez@usj.es).

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
