# Peer review of "Cost-Effectiveness of Two Dry Needling Interventions for Plantar Heel Pain: A Secondary Analysis of an RCT"

_ijerph, 2021, doi:10.3390/ijerph18041777_

Round 1

Reviewer 1 Report

See attached.

Author Response

We have attached a reply point by point in a word document with different colours to make the review process easier

Reviewer 2 Report

I would thank the opportunity to review this manuscript. Unfortunately, I think that there are some aspects of the article that makes it not possible to consider for publication. 

-Since they are two similar treatments, why a cost-effective analysis is performed is not clear. Maybe it should be better explained in the introduction.

-In my opinion, the introduction is very short, it does not explain the literature to show what is known. What are the treatments? I think that authors should include some systematic reviews and meta-analysis to explain the effectiveness of the dry needling in this pathology. 

-The interventions should be better explain. If both groups received stretching, it is difficult to know the effects of each treatment without a control group that receives only the stretching.

-The randomization should be better explained.

-I think that the only use of the EQ-5D to evaluate the effectiveness is very poor. Other articles that evaluate dry needling in PHP used a VAS to evaluate pain, or a tool to evaluate the functionality. Additionally, I think that the medication should be included (especially painkillers).

-Where is the sample size calculation?

Author Response

(The authors gave the same response as above.)

Reviewer 3 Report

The cost-effectiveness analysis of the data gathered from the experimental study with 2 non-invasive interventions was correctly and appropriately executed.  The presentation is well organized and clearly illustrated.  The empirical findings confirm the fact that one intervention is better than the other intervention in the observation of 52 weeks.

The concern is that the authors should clarify if there were any attrition cases during the long period of observation for 52 weeks.  Can additional assessment on the intent-to-treat analysis be conducted to ensure the integrity of experimental outcomes?  In addition, I am not sure if a control group could be established when comparative cost-effectiveness analysis is being performed. 

Author Response

(The authors gave the same response as above.)

Round 2

Reviewer 2 Report

I think that the authors have done significant changes to the article and my recommendation is to accept it in its present form.

Author Response

Here we attach a letter of reply to all reviewers because the system doesn´t allow us to reply one by one like the last time
